# The Influence of the Magnetic Field on DC and the Impulse Breakdown of Noble Gases

**DOI:** 10.3390/ma12050752

**Published:** 2019-03-05

**Authors:** Čedomir I. Belić, Koviljka Đ. Stanković, Milić M. Pejović, Predrag V. Osmokrović

**Affiliations:** 1Faculty of Electrical Engineering, University of Belgrade, 11000 Belgrade, Serbia; cedomirbelic@dmdm.rs (Č.I.B.); kstankovic@vinca.rs (K.Đ.S.); 2Faculty of Electrical Engineering, University of Niš, 18000 Niš, Serbia; milic.pejovic@elfak.ni.ac.rs

**Keywords:** magnetic field, gas breakdown, free electron spectrum

## Abstract

Increased electromagnetic contamination of the environment accompanied with the amplified miniaturization of electronic components underline the issue of the reliable operation of electronics. Reliability is of utmost importance in special applications such as medical instruments, nuclear installations, fusion experiments, etc., where larger magnetic fields occur during operation. Therefore, the interest for insulation components that consistently protect instrumentation from overvoltage is growing. This paper deals with the effects important for the stability of a gas surge arrester, the most commonly used low-voltage component for the overvoltage protection. The effect of the magnetic field on DC and the impulse breakdown of noble gases is investigated. For the theoretical interpretation of the obtained results, the spectrum of free electron gas was determined, which enabled the evaluation of a new expression for the first Townsend coefficient. The results obtained in that way were verified through comparison with theoretically calculated results. Experiments were carried out under well-controlled laboratory conditions.

## 1. Introduction

An increasing degree of miniaturization of electronic components and increased electromagnetic contamination of the environment result in an increase in interest for insulation components at the low-voltage level (low-voltage protection components). The task of these components is to prevent the overvoltage to reach the protected element. For this reason, protective components are connected in parallel to a protected element. The most commonly used low-voltage [1] overvoltage protection component is a gas surge arrester (known as a fuse with a noble gas in German literature) [2,3,4].

The gas surge arrester is a two-electrode system of a homogeneous electric field insulated with noble gas at low pressure. The gas surge arrester works on the principle of the electric gas breakdown. The operating point of the gas arrester is determined by its nominal voltage (DC value of the breakdown voltage) and the product of pressure (p) and a distance between the electrodes (the inter-electrode distance). For the correct design of insulation at the low-voltage level, the stability of the working point of the used surge protection component is extremely important. In the case of a gas surge arrester, the stability of the working point is influenced by its constructive parameters which determine the degree of reversibility of its isolation characteristics after multiple breakdowns. It has been shown that strong magnetic fields stabilize the operating point and decrease the statistical dispersion of the breakdown voltage in the two-electrode system [5,6]. In addition, it has been shown that the environmental parameters in which the surge arrester works can affect the stability of its operating point [6,7].

Among the environmental parameters that significantly affect the stability of the working point of the gas surge arrester, the influence of a magnetic field effect is emphasized. In practice, larger magnetic fields occur in special applications (such as in medical instruments, nuclear installations, fusion experiments, etc.) when the reliable operation of electronics is of the utmost importance. Therefore, the aim of this paper is to examine the magnetic field influence on DC and the impulse breakdown of noble gas [8,9].

## 2. Electrical Probe of Noble Gases in Magnetic Field

At room temperature, each noble gas represents a mixture of neutral atoms, positive ions and free electrons. Assuming that the elastic collision frequency is constant, in case of when inelastic collisions need not be taken into consideration, the formation of a stable electron spectrum of the Maxwell type is thus enabled. This is a consequence of the precise balancing of energy gain in the field and elastic energy loss. However, the distribution function of free electron gas *f*(*ε*) can be further complicated. The situation is not too intangible. Moreover, in case of the large volume filled with weakly ionized monoatomic gases in the weak electrical field (elastic collisions of the electron with neutral atoms and electron with electron are dominant in respect to inelastic collisions, while the large gas volume ensures the diffusion loss to be small), it can be fully applied. More specifically, the described conditions can be applied at the Townsend breakdown of noble gases. In that situation, the elastic collisions of electrons with neutral atoms warrant the density of low energy electrons (up to 10 eV) to be rendered significantly higher than the density of high energy electrons. More precisely, in the range of the Townsend breakdown, electron-electron collisions can be neglected since they become dominant when the degree of ionization reaches values of 10^−3^ and higher. Yet, these values seem to be very far from the Townsend condition of breakdown. This leads to the conclusion that *f*(*ε*) can be considered Maxwellian in the range of the Townsend breakdown. If the distribution function tail decreases with function *C*·exp(−*ε^к^*), where *C* is constant and *k* ≈ 1, this assumption can be considered true [10,11,12]. If the noble gas is subjected to the electric field, the thermal (chaotic) motion of the ionic and electronic components is added to the drift component in the direction of the field. This drift component does not change the shape of their spectrum, but only moves it towards higher energies. This shift towards higher energies is more pronounced in the case of the electron spectrum and may have the effect that the electrons from the tail of the Maxwell spectrum (surpass) exceed the ionization energy of neutral atoms and become initial. Becoming initial means that the free electron takes an energetically favorable state on the free path length and takes enough energy from the electrical field to ionize the neutral Helium atom, creating electron-ion pairs. These newly created electrons can also ionize other neutral Helium atoms and this leads to the avalanche process, which is actually initiated by the very first electron. Hence, that first electron could be called initial. If the avalanches which started by such initial electrons with positive feedback become self-sustaining, there is an electric gas breakdown.

The mathematical modeling of electric gas breakdown is done using the so-called ionization coefficients: *α*, the primary ionization coefficient, and *γ*, the secondary coefficient. In the case of electronegative gases, the coefficient of electron capture, *η*, is also used. The primary ionization coefficient *α* represents the number of electrons generated by ionization processes per unit path length of the initial electron in the direction of the electric field. The secondary coefficient *γ* represents the number of new, free (secondary) electrons per one primary avalanche. The ionization coefficient *α* does not have a constant value, but it changes depending on the ratio of the electric field and pressure (which is in accordance with the law of similarity for electric discharge in gases). The secondary coefficient *γ* depends on many elements and changes in the range of 10^−3^ to 10^−8^ [13,14,15].

The dependence of the primary ionization coefficient *α* on the ratio of the electric field and gas pressure is routinely determined by the semiempirical method. The variation of the gas atoms density (*n*) over a distance (*x*) is described by the exponential differential equation *dn = α(x)∙n∙dx*. The integration constants are determined on the basis of the experimental results. The obtained dependences are applicable only within a certain interval of the electric field vs pressure ratio values. In practice, the Townsend or the Takashi ionization coefficient terms are mostly used. However, the knowledge of the shape of the free electron energy spectrum allows the primary ionization coefficient to be determined exactly by definition [16]:(1)$$\alpha \left(x\right)=n_{0}\int_{\varepsilon_{i}}^{\infty }\sigma_{i}\left(\varepsilon \right)vf\left(\varepsilon \right)d\varepsilon$$ where *v* is the free electron velocity in the direction of the electric field, *ε* is the free electron energy, *σ_i_* (*ε*) is the effective ionization cross section (which is practically independent of the free electron energy), *n*_0_ is the density of the neutral gas atoms (which is proportional to the pressure of the gas), and *f*(*ε*) is the energy distribution of the free electrons [14,15].

Replacing the expression for Maxwell distribution
(2)$$f\left(\varepsilon \right)d\varepsilon =2\sqrt{\frac{\varepsilon }{\pi }}{\left(\frac{1}{kT}\right)}^{\frac{3}{2}}\cdot \exp\left(-\frac{\varepsilon }{kT}\right)d\varepsilon$$ in Equation (1), gives us the following:(3)$$\alpha \left(T_{e}\right)=4\frac{M\sigma_{i0}}{R\sqrt{\pi }}\cdot p\cdot \frac{\varepsilon_{i}+2T_{e}}{T_{e}}\cdot \exp\left(-\frac{\varepsilon_{i}}{T_{e}}\right)$$ where *p* is the gas pressure, *M* is the gas molar mass, *σ*_*i*0_ is the effective ionization cross section of the neutral gas molecules by electron energy *ε_i_*, *R* is the Rydberg constant and *T_e_* is the temperature of free electrons gas. The temperature of the free electron gas is determined by the expression [10]
(4)$$T_{e}=kT=\xi \lambda eE=\xi \lambda e\frac{U}{d}$$ where *ξ* is the thermalization from factor, *U* is the applied voltage, *d* is the inter-electrode distance, and *e* is the elementary charge (1.6 × 10^−19^ C). It should be noted that the second part of the equation (expression *ξλeE* in) Equation (4) applies only in the case of a homogeneous or a pseudo-homogeneous electric field. The mean energy of the free electron gas spectrum is obtained on the basis of Maxwell’s distribution and the electronic energy balance [10]:(5)$$\bar{\varepsilon }=0.8eE\frac{\lambda }{\sqrt{\delta }}$$ where *E* is the electric field, *λ* is the mean free path length and *δ* is the ratio of the electron mass and the molar mass of the gas.

When a noble gas is found in the electric field, the thermal movement speed of its charged components is superposed by one equally accelerated component in the direction of the field. If free electrons accelerating in the direction of the field acquire energy that is higher (or equal to) than the neutral gas ionization energy at one mean free path length of the path, they become initial. The initiated electron avalanche may eventually trigger a self-sustaining mechanism. Established self-sustaining discharge is controlled by the relationship of the collisions of charged particles, as well as their transport properties. Depending on the self-discharge mechanism, the electrical breakdown can be performed by the Townsend or streamer mechanism.

If the secondary processes that are active on electrodes (ion discharge, photoemission or metastable discharge) dominate over the secondary processes that are active in the gas (ionization by positive ions, photoionization, metastable ionization), the breakdown takes place with the Townsend mechanism. This occurs in the case where the characteristic dimensions of the insulation system are comparable to the mean free path length of the electron [17,18]. The condition for Townsend’s breakdown of an inhomogeneous field is
(6)$$\gamma \int_{0}^{d}\alpha \left(x\right)\cdot \exp\left(\int_{0}^{x}\alpha \left(x\right)dx\right)dx=1$$

If the secondary processes that are active in the gas dominate over the secondary processes that are active on electrodes, the breakdown takes place with the streamer mechanism. This occurs in the case where the characteristic dimensions of the insulation system are significantly greater than the mean free path length of the electron [19,20]. The condition for a streamer breakdown in a non-homogeneous electric field is
(7)$$\int_{0}^{d}\alpha \left(x\right)dx=10.5$$

Given the fact that neither the Townsend nor the streamer mechanism has the inclination to be exclusive, establishing a straightforward boundary between these two mechanisms is not advisable. As a consequence, it is assumed that a broader range of PD product values can be expected to bring about breakdowns, thereby combining the two mechanisms. The results of examining the influence exerted by the electrode material and treatment of the electrode surface on the static voltage breakdown value lend support to such a supposition. It could be considered that if the values of the inter-electrode gap and the mean path length of the free electron are approximately comparable, the Townsend mechanism of breakdown is dominant. If the inter-electrode gap length is much higher than the mean path length of the free electron, the streamer mechanism occurs. However, it can be assumed that the breakdown actually occurs within a certain boundary region which is formed. The breakdown is initiated by the Townsend mechanism and maintained by the streamer.

The previously described conditions for the Townsend and streamer breakdown mechanism of the noble gases imply the existence of an electric field in an inter-electrode space, from which a free electron can, within a mean free path length, achieve energy equal to the ionization energy of neutral atoms. However, nothing was mentioned about the changing rate of that electric field. Considering the change rate of the electric field, Equations (6) and (7) describe the conditions for a gas breakdown using an electric field which has an increasing rate that is much lower than the time constant of the elementary processes of the electrical discharge in gases. Such an electrical breakdown is called a static or DC breakdown. It is generated by a slowly growing potential difference between electrodes. The DC breakdown voltage is the deterministic value and its associated measurement uncertainty type A is 0 [21].

In case the increased time of a potential difference between the electrodes is of the same order of magnitude as the time constants of the elementary processes of electrical discharge in gases, a dynamic or impulse breakdown occurs. This type of breakdown is achieved by the impulse voltage, which has the double-exponential form, according to the standard [22]. The impulse breakdown voltage is a stochastic value and it belongs to the measurement uncertainty type A > 0. For this reason, the gas insulation structures in relation to the impulse voltage load are probabilistically characterized. Most often, the semiempirical surface law (Appendix A) determines the voltage-time area, where all data points of the breakdown voltage-breakdown time are located with a known probability. The boundaries of this domain are usually referred to as impulse characteristics [23,24,25].

When the charged particle with the charge q is in the magnetic field, the Lorentz force that acts on it is defined by the expression
(8)$$F=q\;\vec{V}x\vec{B}$$ where $\vec{V}$ is the particle velocity vector, and $\vec{B}$ is the magnetic induction vector (Appendix B). Due to the Lorentz force, charged particles rotate around the magnetic induction vector $\vec{B}$ along the circle with a radius of
(9)$$r=\frac{m_{0}}{q}\frac{V_{m}}{B}$$ where $m_{0}$ is the mass of the charged particle and $V_{m}$ is the component of the charged particle velocity normal to the magnetic induction vector. If the charged particle moves in the electromagnetic field, the resultant force is described by the Lorentz equation:(10)$$\vec{F}=q\left(\vec{E}+\vec{V}x\vec{B}\right)$$ where $\vec{E}$ is the electric field vector. The effect of the electromagnetic field leads to the spiral motion of the charged particles showed in Figure 1.

## 3. Experimental Setup

As mentioned above, the aim of the experiment was to examine and explain the magnetic field effect on the DC and impulse breakdown voltage. In order to do this, it was necessary to first determine the free-electron gas spectrum.

The free-electron gas spectrum was determined on the basis of 1000 measurement values of the impulse breakdown voltage obtained by impulses of 1 kV/s. It was assumed that there is a fraction of thermal electrons needing a small amount of energy, taken from the electric field, to become an initial one (i.e., free electrons which acquire enough energy after a few mean free path lengths to carry out the ionization process of a neutral gas atom). Such electrons form a high energy tail in the spectrum. On the histogram of the breakdown voltage values, these electrons belong to the lowest breakdown values. This allows drawing a histogram of the high energy tail spectrum of free electrons. In order to determine whether this tail histogram really belongs to the Maxwell or the Druyvesteyn distribution, additional testing was carried out [10]. In all examined cases, it was observed that the free electron gas spectrum belongs to the Maxwell distribution (i.e., the tail of the energy spectrum decreases proportionally to the exponent energy to the first degree, and not to the second degree that would correspond to the Druyvesteyn distribution). After this conclusion, it was possible to fit (using Equation (2)) the entire histogram of the impulse breakdown voltage values and obtain an analytical expression for the Maxwell distribution of the free electron gas spectrum.

After determining the Maxwell distribution, it was possible to calculate the dependence of the ionization coefficient α (E/p) from Equation (3). Knowing the ionization coefficients α (E/p) and γ allowed for plotting the theoretical Paschen’s curve. The theoretical Paschen’s curve is then compared with the experimental curve, which was obtained with the hollow cathode and the source ^241^Am in it. This comparison provides verification of the method previously used to determine the Maxwell distribution of free electron gas. Knowing the parameters of the Maxwell spectrum of free electrons allows the interpretation of the obtained results related to the influence of the magnetic field in the inter-electrode space on the *DC* breakdown voltage.

The illustration of the two gas chambers used in the experiment is shown in Figure 2. One chamber was designed to operate in the underpressure regime/mode (pressure in the chamber) and the other in the overpressure mode (pressure in the chamber). The difference between these two chambers was in the *O-ring* channel profile. The electrodes of the axially symmetric system made of copper were in the form of the Rogowski (to avoid the electric field edge effect). For each inter-electrode distance, a new pair of electrodes was made (using the calculation of the electric field lines obtained by the charge simulation method). In one of the electrodes (commonly used as a cathode) it was possible to install 1, 2 or 3 identical cylindrical Alnico magnets of induction Br = 1,2 T (cylindrical magnets with the strongest induction value that was available for the experiment), with a radius of R = 10 mm and a height of N = 10 mm (shown as detail in Figure 2). These magnets created a magnetic field collinear with the electric field in the inter-electrode space, with an induction of 0.58 T on the surface of the cathode (Appendix C). Before each series of measurements, the electrodes were sanded. An electrode (cathode) with an axial cavity containing radioactive ^241^Am was also used. This electrode was used to obtain the experimental points in the region of Paschen’s curve, that is, on the left side of Paschen’s minimum (i.e., without the appearance of an anomalous Paschen’s effect) [16].

Adjustment of the inter-electrode distance was carried out by electrode motion in three steps: 1) determination of the zero distance by measuring the ohmic resistance; 2) determining the inter-electrode distance by means of a fine thread of the electrode carrier and an electronic micrometer and 3) fixing the position of the electrode by a counter-screwing. The Type B measurement uncertainty of the inter-electrode distance adjustment in this procedure was 0.2%. This result is confirmed by measuring 1000 values of DC breakdown voltages (the Type A measurement uncertainty was zero).

The experiments were carried out with the noble gas He. Figure 3 shows a gas circuit for adjusting the pressure value in the chamber. The pressure adjustment procedure was performed in the following steps: 1) vacuuming the chamber to a pressure of 1 Pa; 2) gas injection to a pressure of 10^5^ Pa; 3) repeat steps 1 and 2 twice and 4) reduce the desired pressure value of the chamber to 0° C by using Gay-Lussac’s law.

Figure 4 shows the scheme of the instrumentation used in the experiment. A *DC* voltage with a rise rate of 8 V/s was used, as well as impulse voltages with rise rates of 1 kV/s and 1 kV/μs (amplitudes much larger than the possible value of the impulse breakdown voltage). The chamber (cathode) was grounded using 1 MΩ of secured negligible irreversible changes in the topography of the electrode surface during measurement. The success of this measure was confirmed by Talysurf before and after one series of measurements. By examining the electrode surfaces by microscope, it was concluded that the breakdown points were uniformly distributed along the electrode surfaces. The measurement cabin was not in galvanic contact with the rest of the experimental set-up. A voltage divider and digital voltmeter were used to measure the *DC* voltage. For the measurement of the impulse voltage, a compensated capacitive voltage divider and a 500 MHz digital oscilloscope were used. The process of voltage sources control and the acquisition of the measured values were completely automated. Following the electrode configuration of the chamber, the gas chamber was plugged into a circuit. For one pressure value, after conditioning the electrode system, the following measurements were performed: 1) 50 values of the *DC* breakdown voltage; 2) 1000 values of the impulse breakdown voltage obtained by an impulse of 1 kV/s; 3) 100 values of the impulse breakdown voltage obtained by an impulse of 1 kV/μs; 4) repeating the previous measurements for a new pressure value; 5) adjusting the inter-electrode distance and repeat steps 1 to 4; 6) placing one form of magnet in the cathode and repeating steps 1 to 5; 7) placing two magnets in the cathode and repeating steps 1 to 5; and 8) placing three magnets in the cathode and repeating steps 1 to 5. A pause of 30 s was made between the two successive steps. The values of the inter-electrode distance were 0.1 mm, 0.5 mm and 1 mm. The pressure values were changed from 5000 Pa to 10,000 Pa with a step of 500 Pa, and also from 50,000 Pa to 150,000 Pa with a step of 10,000 Pa. At the points on the left side of Paschen’s minimum, additional *DC* breakdown voltage measurements were made with a smaller difference in the pressure variation. Using the Monte Carlo statistical procedure and analytical procedures, it was found that the combined measurement uncertainty of the procedure was less than 5% [26].

On the basis of the results obtained by measuring the impulse voltage of 1 kV/s, the parameters of the free electron gas spectrum were determined. The pressure, the inter-electrode distance and the magnetic induction value in the inter-electrode space were the measurement parameters. The obtained results were compared with the results of the numerical and theoretical considerations.

Based on the results obtained by measuring the *DC* voltage, the values of the *DC* breakdown voltage versus the pressure ratio were determined. The inter-electrode distance was the measurement parameter. Additionally, the dependence of the *DC* breakdown voltage versus the product *PD*, the so-called Paschen’s curve, is determined for two cases: with and without a hollow cathode containing a radioactive source. In the case without a hollow cathode, statistical samples of the DC breakdown voltage random variables were divided into the sub-samples. The division was performed in a way which guarantees that all random variables in the sub-samples belong to the same statistical sample with a statistical uncertainty of less than 1%. This division was carried out at the points on the left side of Paschen’s minimum since, in the points on the right side, the required condition was fulfilled for the entire statistical sample (i.e., the statistical deviation in those points was about 0). All obtained results are compared with the results of the numerical and theoretical calculation.

The impulse characteristics were determined from the results obtained in the measurements using an impulse voltage with a rise rate of 1 kV/μs, and by application of the surface law. The pressure, inter-electrode distance and magnetic induction value in the inter-electrode space were the measurement parameters. The experimental procedure can be described by the flowchart shown in Figure 5.

## 4. Results and Discussion

The free-electron gas spectrum was observed based on 1000 measurements using slow impulse voltage. More accurate results can be obtained using DC voltage, but in that case, the experiment would last much longer. The idea of the experiment was to record the impulse breakdown voltage probability histogram. From that histogram, a small probability of the breakdown at small voltages was extracted. A small probability of the breakdown was achieved by low voltage values [1], microscopically corresponding to the electrons from the tail in the energy spectrum. The electrons from the tail could take only a small amount of energy from the electric field (during one free path length) to perform ionization. Hence, by knowing the mean path length of the free electron and ionization energies of Helium, it is possible to draw a tail of the free-electron gas energy histogram. Figure 6a, Figure 7a and Figure 8a show histograms of class 1 + 3.3•log(n) (n is the size of a statistical sample) of a free electron gas energy distribution and were obtained by fitting the distribution tail with expression *C*∙exp(−*ε^к^*). By fitting the tail of the histogram, the value of fitting parameter *k ≈* 1 was obtained. Based on that result, it was concluded that the histogram distribution is Maxwellian and the tail of the histogram could be fit using Equation (2). The complete Maxwellian distribution of free electrons is shown in Figure 6b, Figure 7b and Figure 8b. Table 1 gives the values of the mean energy of the free electron gas spectrum based on the results shown in Figure 6, Figure 7 and Figure 8 and Equation (5).

Based on results (Figure 6, Figure 7, Figure 8 and Table 1) it can be concluded that the free electron gas spectrum in the noble gas helium (He) is of the Maxwellian shape. Additionally, the overall energy of the initial electron (which increased for the amount of rotation energy around vector *B*) does not affect the Maxwellian shape of the free electron gas spectrum. The latter is due to the existence of an elastic type of collision between the He atoms and free electrons. Such a state of thermodynamic equilibrium is neither disturbed by an electric nor a magnetic field. A smaller deviation of the free electron gas spectrum occurs at higher pressure values as a result of the appearance of the inelastic Coulomb electron-electron interactions. Although shown in Figure 6, Figure 7 and Figure 8 and Table 1, this phenomenon does not significantly disturb a shape of the free electron gas spectrum (it only leads to an increase in the spectrum density at lower energies).

As already stated, the cross-section *σ_i_(ε)* is practically independent of the free electron energy and *v* is the velocity in the direction of the electric field independent from magnetic field B. In addition, and consistent with the results given in Figure 6, Figure 7 and Figure 8, the shape of the Maxwellian spectrum does not depend on the presence of the magnetic field in the inter-electrode gap. Thus, Equation (1) allows for the determination of the ionization coefficient. This means that the magnetic field has a negligible influence on the ionization (avalanche) process, but does affect the value of the breakdown voltage because the superposition of translational and rotational motion of the initial electron leads to an increase of its path length i.e., to an apparent increase in the inter-electrode distance (Appendix B).

These conclusions are confirmed by the results shown in Figure 9. The results show that the experimentally obtained values of the *DC* breakdown voltage (using a hollow cathode with a source of ^241^Am) depend on the product *PD* (pressure × inter-electrode distance) together with the theoretical dependence obtained by Equations (3), (4), (6) and (7). The results show a good agreement between the experimentally obtained and theoretically (analytically) calculated results. This proves that the expression for the primary ionizing coefficient α derived under the assumption of the Maxwell distribution of the free electron gas spectrum in a noble gas is applicable to a large range of the values of the E/p coefficients (i.e., the product *PD*). A small deviation between the experimental and theoretically determined values of the DC breakdown voltages is observed at higher pressures (Figure 9). That is, due to the inelastic electron-electron Coulomb interactions at higher gas densities, the effect that was pointed out previously.

Figure 10, Figure 11, Figure 12, Figure 13, Figure 14, and Figure 15 show the experimental points of the *DC* breakdown voltage versus the pressure of the gas (He). Experiments were performed with three cylindrical Alnico magnets in the cathode and with an inter-electrode distance of 0.5 mm. Experimental points are fitted using the Paschen’s curve with the inter-electrode distance as a parameter in the case of the existence of a magnetic field in the inter-electrode space. The same figures also show the theoretical Paschen’s curves obtained without a magnetic field in the inter-electrode space. The captions of the figures give the values of the best fit and the values calculated using Equations (6), (7) and (A16) (Appendix B—the parameters necessary for the calculation of the Equation (A16) were taken from the predetermined Maxwell distribution of the free electron gas under the experimental conditions).

The results above show that the effect of the magnetic field in the inter-electrode space leads to a noticeable inter-electrode distance expansion. Consequently, the values of the DC breakdown voltage at the points to the left of Paschen minimum (Figure 10, Figure 11 and Figure 12) are lower when there exists a magnetic field in the inter-electrode gap and vice versa, at the points to the right of the Paschen minimum (Figure 13, Figure 14 and Figure 15). The latter is attributable to the anomalous Paschen effect, which appears at the points to the left of the Paschen minimum [27]. According to the anomalous Paschen effect, lower values of the breakdown voltage correspond to higher inter-electrode distances. This effect decreases with increasing magnetic field values, which is in accordance with Equation (A16). The quantitative results of the apparent spread of the inter-electrode distance, obtained experimentally, are greater by about 10% to 15% of the expected results based on Equation (A16). This difference is probably due to a sudden decrease in the magnetic field value in the inter-electrode space (Figure A3).

This evident inter-electrode distance expansion is due to the spiral motion of the electron. However, due to the different angles between the rotational planes of the free electron gas components and the electric field for a large number of spirals, breakdown is possible. At points on the right side of the Paschen’s minimum, the breakdown will always occur along the spiral that allows for the shortest path. However, at points on the left side of the Paschen’s minimum, the breakdown may happen along the longer spiral if it is more energy efficient. The existence of a large number of different spirals should cause a phenomenon known as the anomalous Paschen’s effect on the points on the left side of Paschen’s minimum, which is more pronounced in the case when the magnetic field exists in the inter-electrode space. This assumption is confirmed by the results shown in Figure 16.

Experiments with an impulse voltage of 1.2/50 μs have shown that the magnetic field in the inter-electrode space leads to an increase in the mean value of the impulse breakdown voltage, but also to a reduction in its statistical dispersion. This is best illustrated by the impulse characteristics. Figure 17 shows the impulse characteristics determined by the surface law with and without a magnetic field in the inter-electrode space.

The increase in the mean value of the impulse breakdown voltage is due to the apparent increase of the inter-electrode distance under the action of the magnetic field. The decrease in statistical dispersion occurs probably due to the rotational motion of the electron, which keeps the electron longer in a critical volume and thereby increase the probability of becoming the initial one.

## 5. Conclusions

On the basis of the obtained results, which were experimentally confirmed and theoretically explained, the following can be concluded:The free electron gas spectrum in the noble gas is of the Maxwell type.A smaller deviation of the Maxwell shape of a free electron gas spectrum in the noble gas at higher pressures is due to inelastic Coulomb electron–electron interactions.Based on the knowledge of the free electron gas spectrum, it is possible to determine the dependence of the primary ionization coefficient on the ratio between the electric field and the pressure. The primary ionization coefficient obtained in this way is applicable to a wider range of the electric field and pressure values than it is defined in the expressions of the Townsend and Takeshi.The magnetic field, which is collinear with an electric field in the inter-electrode space, leads to an apparent increase in the inter-electrode distance. This leads to an increase in the mean values of *DC* and the impulse breakdown voltage. Due to the existence of a magnetic field in the inter-electrode space, the statistical dispersion of the impulse breakdown voltage value decreases and the anomalous Paschen effect is more pronounced.Gas surge arresters (or similar components such as GM counters) in operating conditions where magnetic field occurs may become ineffective due to an increase in the level of protection (i.e., a change in the nominal operating voltage). The apparent increase of the inter-electrode distance increases the breakdown voltage (nominal voltage). Increasing the breakdown voltage leads to the shifting of the operating point of the GFSA towards higher (U, pd) values. That effect can jeopardize the efficiency of the insulation at a low voltage level.

## Figures and Tables

**Figure 1 materials-12-00752-f001:**
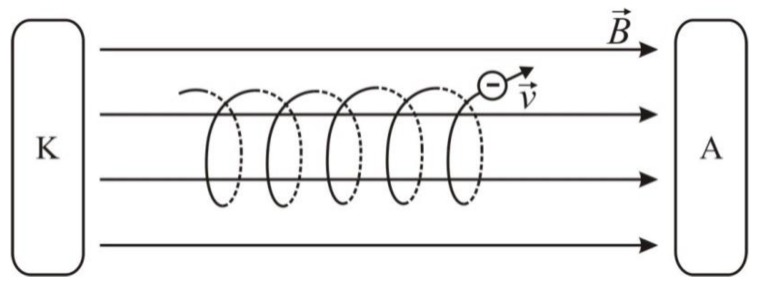
The motion of electrons in electromagnetic fields; K: cathode, A: anode.

**Figure 2 materials-12-00752-f002:**
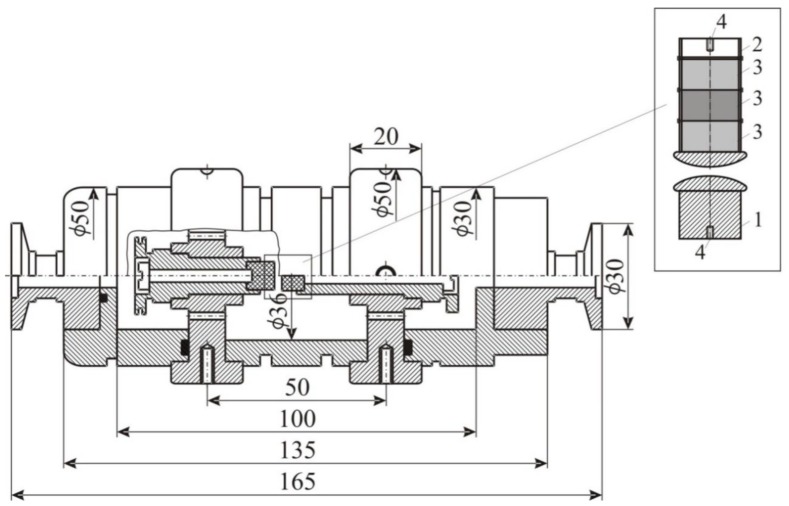
The cross-section of the gas chamber and cross-section of the electrode system; **1**: anode, **2**: cathode, **3**: Alnico magnets, **4**: thread. The units are in mm.

**Figure 3 materials-12-00752-f003:**
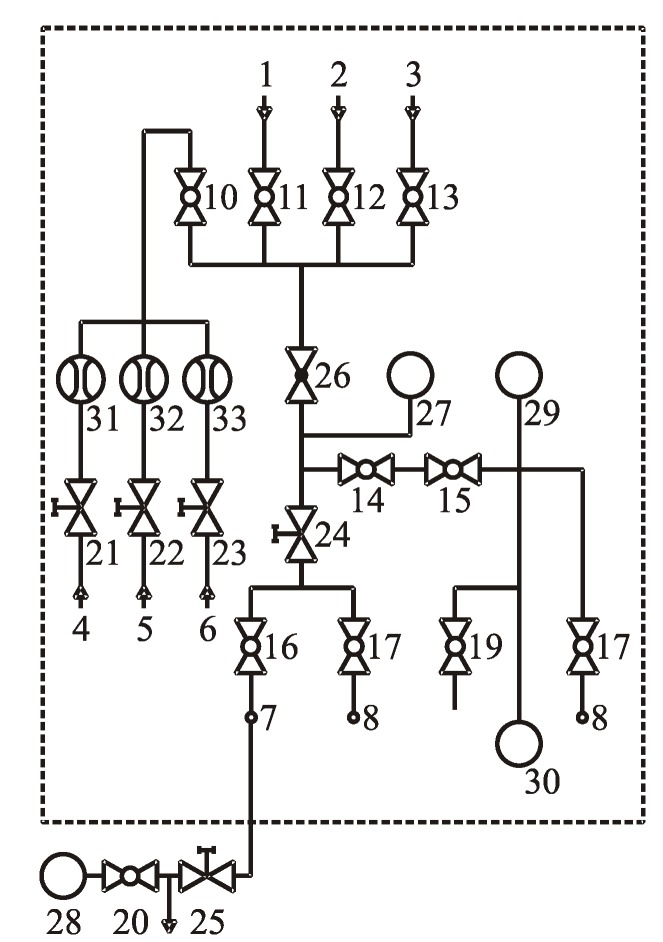
The gas piping scheme (**1**–**9**: gas flows, **10**–**20**: dual-position valves, **21**–**25**: dosing valves, **26**: pressure reducers, **27**–**29**: vacuum gauge, **30**: vacuum pumps, **31**–**33**: relative pressure gauge).

**Figure 4 materials-12-00752-f004:**
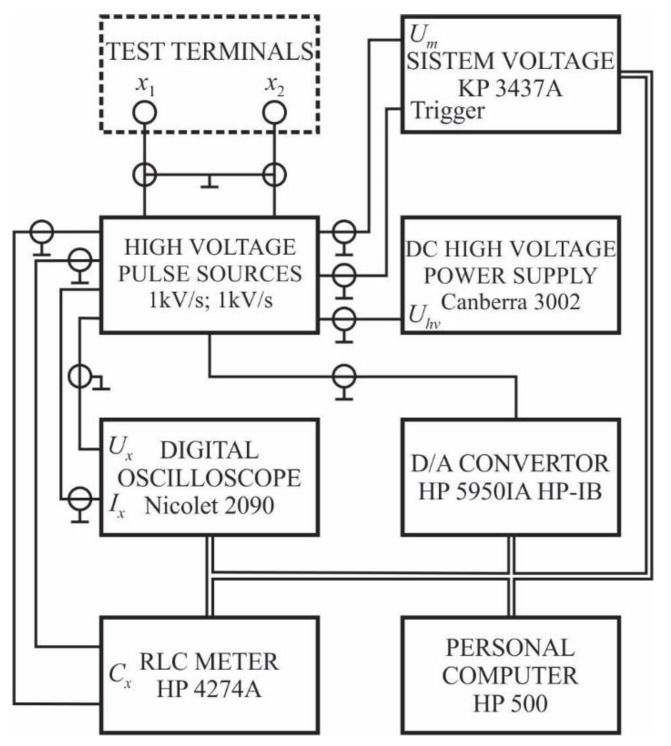
The scheme of the experimental circle used.

**Figure 5 materials-12-00752-f005:**
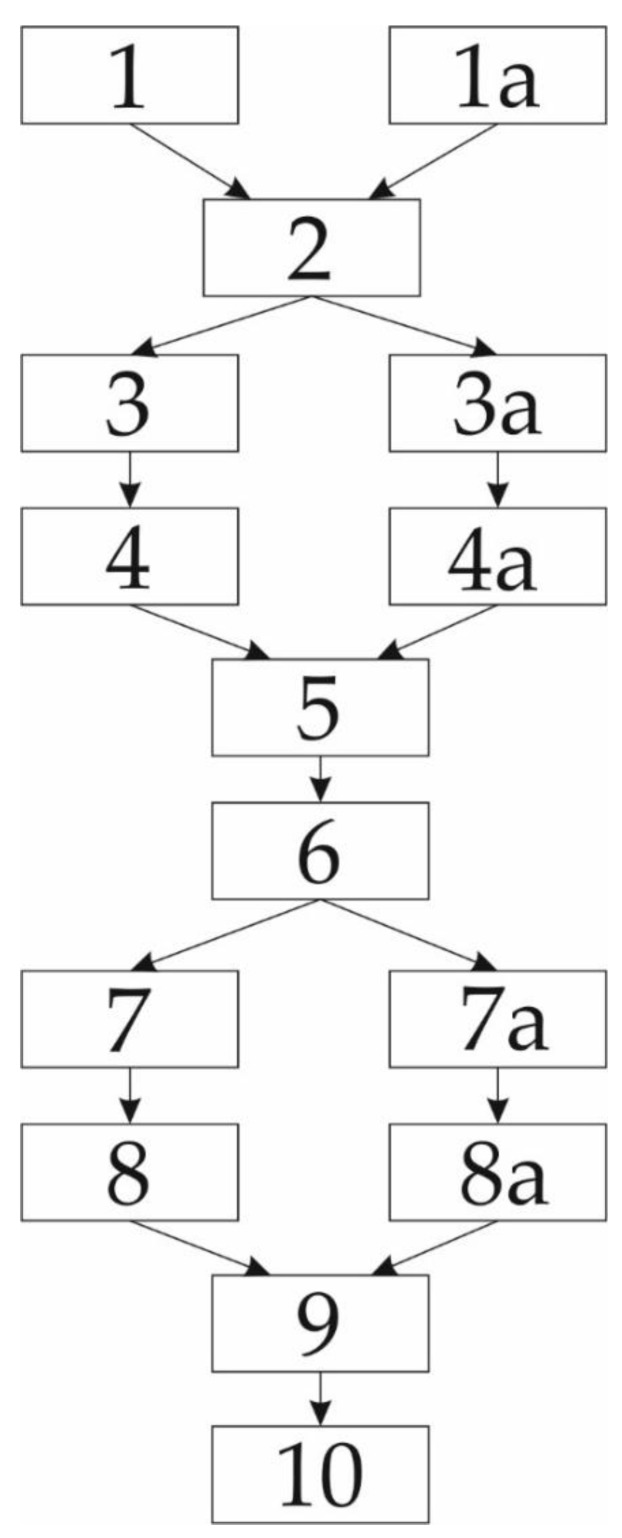
The experimental procedure flowchart; (**1**) adjustment of the chamber operating point without cylindrical magnets in the electrodes; (**1a**) adjustment of the chamber operating point with cylindrical magnets in the electrodes; (**2**) chamber conditioning; (**3**) measuring 1000 values of the impulse breakdown voltage values using 1 kV/s impulses without cylindrical magnets in the electrodes; (**3a**) measuring 1000 values of the impulse breakdown voltage values using 1 kV/s impulses with cylindrical magnets in the electrodes; (**4**) determination of the high energy tail distribution of the free-electron gas without cylindrical magnets in the electrodes; (**4a**) determination of the high energy tail distribution of the free-electron gas with cylindrical magnets in the electrodes; (**5**) fitting the high energy tail (with and without cylindrical magnets in electrodes) and drawing the obtained results on the unique diagram; (**6**) determination of the ionization coefficients; (**7**) measuring the DC breakdown voltage values without cylindrical magnets in the electrodes; (**7a**) measuring the DC breakdown voltage values with cylindrical magnets in the electrodes; (**8**) drawing Paschen’s curves based on the obtained values for α ionization coefficients and drawing (on the same diagram) the measured DC breakdown voltage values (all without cylindrical magnets in the electrodes); (**8a**) fitting the measurement results of the DC breakdown voltage versus the PD values by using the general form of Paschen’s curve; (**9**) the impulse characteristics were determined from the results obtained in the measurements; (**10**) merge the observed the results from step 8 and 8a in a unique diagram.

**Figure 6 materials-12-00752-f006:**
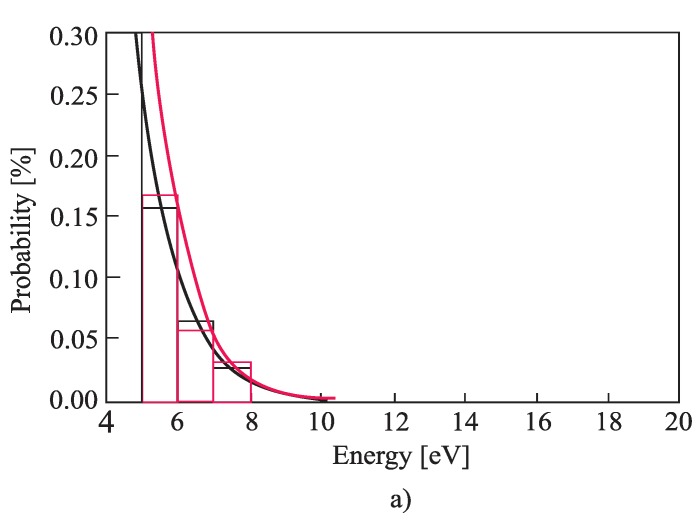
The histogram of class 1 + 3.3·logn of a free electron gas energy distribution; (**a**) the tail of the spectrum for k = 0.96; marked black: without cylindrical magnets in the cathode; marked red: with cylindrical magnets in the cathode; (**b**) the complete spectrum and the Maxwell distribution; p = 5 Pam, p = 5000 Pa; marked black: without cylindrical magnets in the cathode; marked red: with cylindrical magnets in the cathode; Gas: He.

**Figure 7 materials-12-00752-f007:**
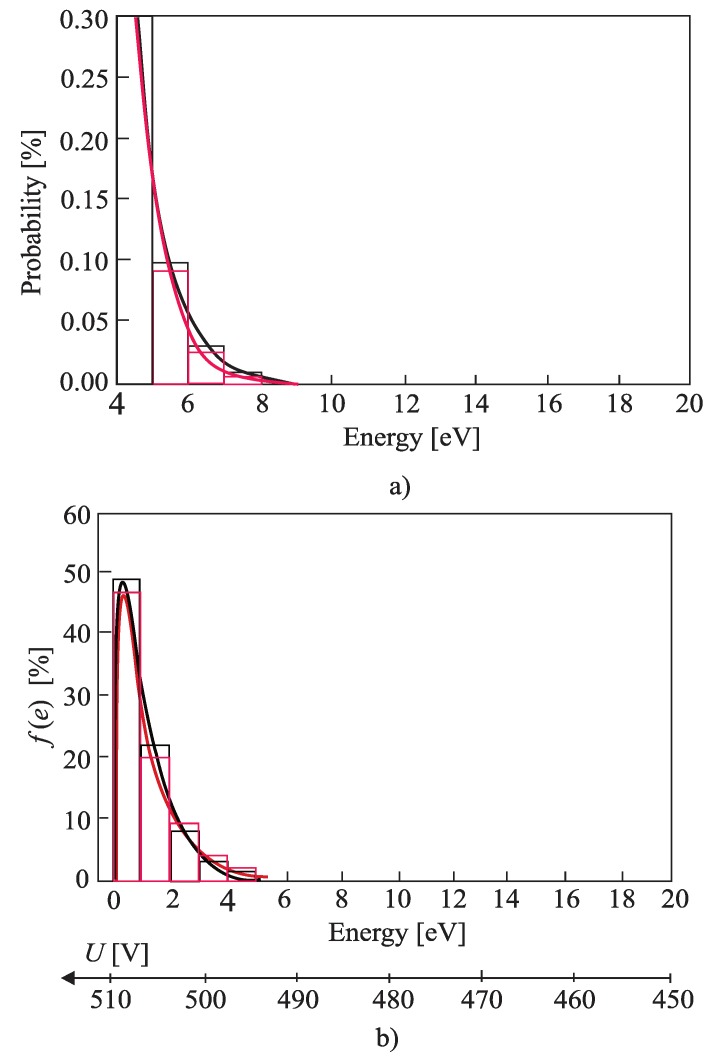
The histogram of class 1 + 3.3·logn of a free electron gas energy distribution; (**a**) the tail of the spectrum for k = 1.05; marked black: without cylindrical magnets in the cathode; marked red: with cylindrical magnets in the cathode; (**b**) the complete spectrum and the Maxwell distribution; p = 5 Pam, p = 50,000 Pa; marked black: without cylindrical magnets in the cathode; marked red: with cylindrical magnets in the cathode; Gas: He.

**Figure 8 materials-12-00752-f008:**
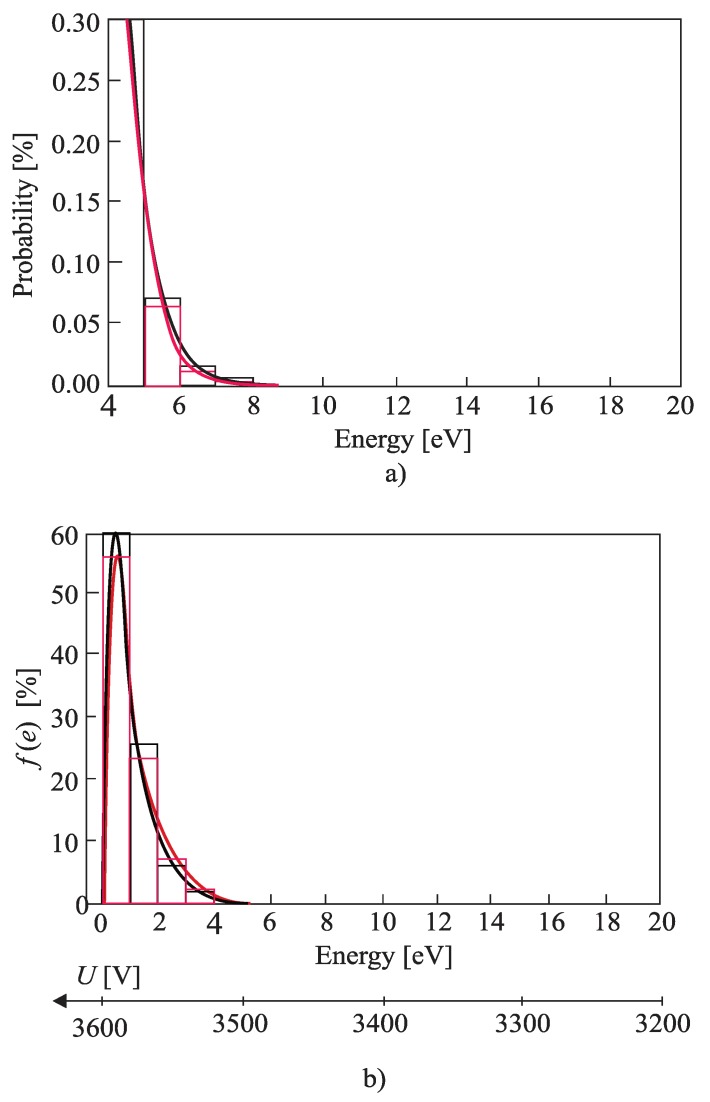
The histogram of class 1 + 3.3·logn of a free electron gas energy distribution; (**a**) the tail of the spectrum for k = 1.16; marked black: without cylindrical magnets in the cathode; marked red: with cylindrical magnets in the cathode; (**b**) the complete spectrum and the Maxwell distribution; p = 105 Pam, p = 105,000 Pa; marked black: without cylindrical magnets in the cathode; marked red: with cylindrical magnets in the cathode; Gas: He.

**Figure 9 materials-12-00752-f009:**
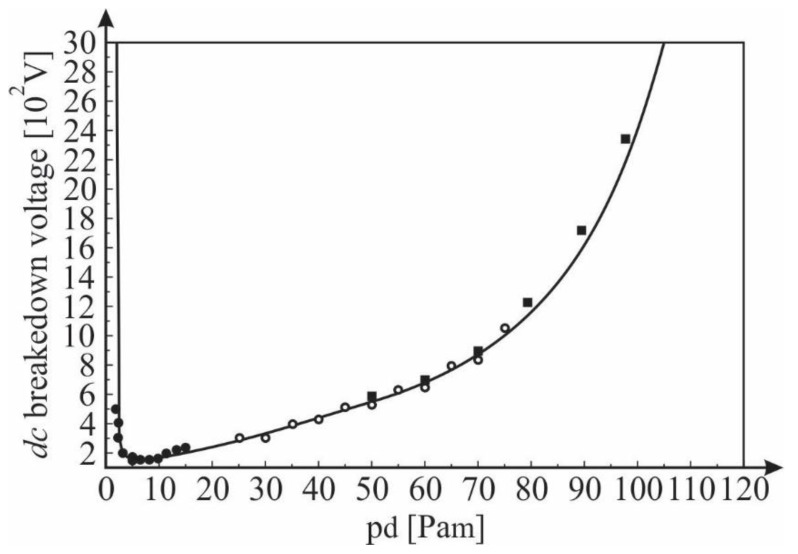
The theoretically calculated *DC* breakdown voltage versus *PD* and values obtained experimentally using a hollow cathode with a ^241^Am source and various *d* (● 0.1 mm, ○ 0.5 mm, ■ 1 mm); Gas: He.

**Figure 10 materials-12-00752-f010:**
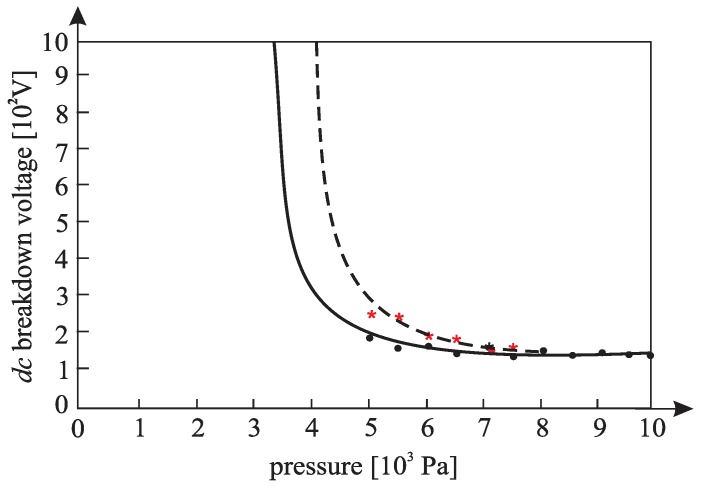
The experimental points (●—: the *DC* breakdown voltage, pressure) fitted using Paschen’s curve with inter-electrode distance as the fitting parameter with cylindrical magnets in the cathode; the experimental points (★---- : the *DC* breakdown voltage, pressure) fitted using Paschen’s curve with an inter-electrode distance as the fitting parameter without cylindrical magnets in the cathode; an experiment performed with three cylindrical Alnico magnets in the cathode, an inter-electrode distance of 0.5 mm; the best fitting parameter is d = 0.61 mm; the value calculated by Equation (A16) is d = 0.58 mm; Gas: He; the magnetic induction was 0.58 T on the surface of the cathode.

**Figure 11 materials-12-00752-f011:**
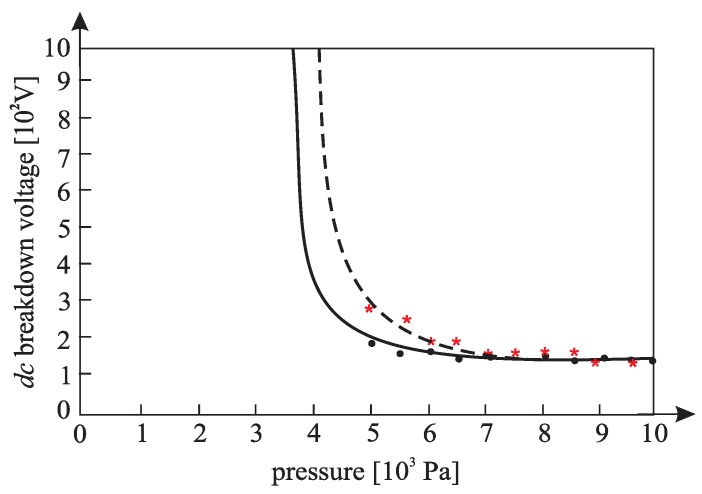
The experimental points (●—: the *DC* breakdown voltage, pressure) fitted using Paschen’s curve with inter-electrode distance as the fitting parameter with cylindrical magnets in the cathode; the experimental points (★---- : the *DC* breakdown voltage, pressure) fitted using Paschen’s curve with inter-electrode distance as the fitting parameter without cylindrical magnets in the cathode; an experiment performed with three cylindrical Alnico magnets in the cathode, an inter-electrode distance of 0.5 mm; a best fitting parameter of d = 0.58 mm; the value calculated by Equation (A16) was d = 0.56 mm; Gas: He; the magnetic induction was 0.58 T on the surface of the cathode.

**Figure 12 materials-12-00752-f012:**
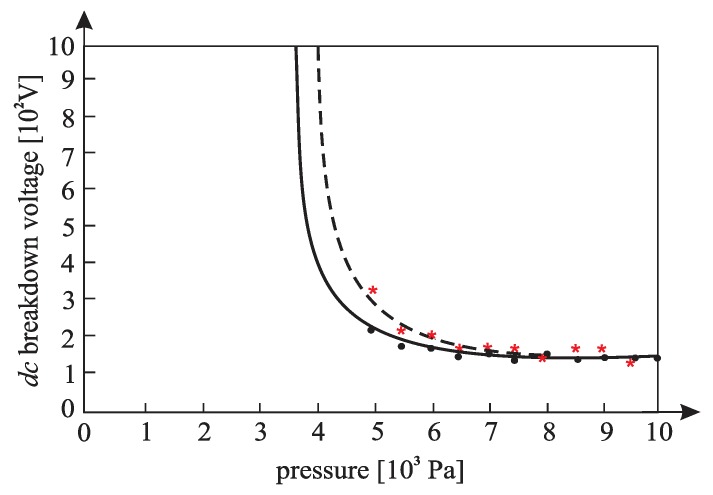
The experimental points (●— : *DC* breakdown voltage, pressure) fitted using Paschen’s curve with inter-electrode distance as the fitting parameter with cylindrical magnets in the cathode; the experimental points (★---- : *DC* breakdown voltage, pressure) fitted using Paschen’s curve with inter-electrode distance as the fitting parameter without cylindrical magnets in the cathode; an experiment performed with three cylindrical Alnico magnets in the cathode, an inter-electrode distance of 0.5 mm; a best fitting parameter of d = 0.56 mm; the value calculated by Equation (A16) was d = 0.53 mm; Gas: He; the magnetic induction was 0.58 T on the surface of the cathode.

**Figure 13 materials-12-00752-f013:**
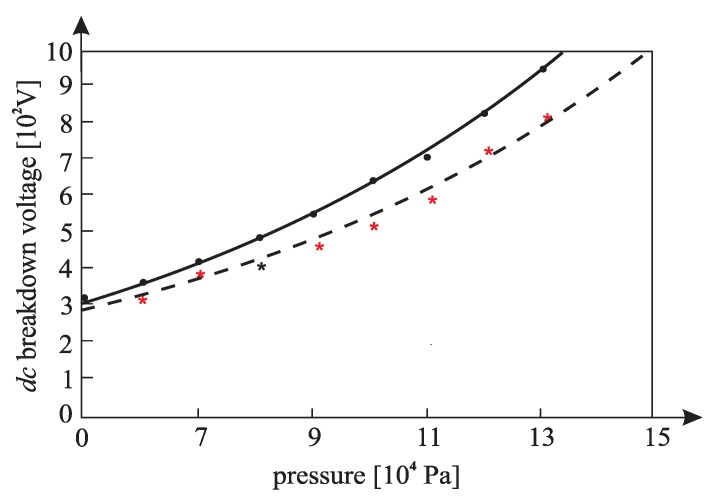
The experimental points (●— : *DC* breakdown voltage, pressure) fitted using Paschen’s curve with inter-electrode distance as the fitting parameter with cylindrical magnets in the cathode; the experimental points (★---- : *DC* breakdown voltage, pressure) fitted using Paschen’s curve with inter-electrode distance as the fitting parameter without cylindrical magnets in the cathode; an experiment performed with three cylindrical Alnico magnets in the cathode, an inter-electrode distance of 0.5 mm; a best fitting parameter of d = 0.59 mm; the value calculated by Equation (A16) was d = 0.57 mm; Gas: He; the magnetic induction was 0.58 T on the surface of the cathode.

**Figure 14 materials-12-00752-f014:**
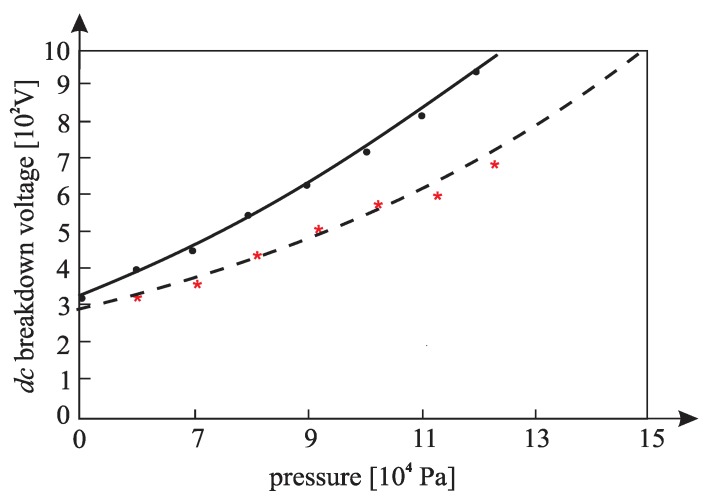
The experimental points (●— : *DC* breakdown voltage, pressure) fitted using Paschen’s curve with inter-electrode distance as the fitting parameter with cylindrical magnets in the cathode; the experimental points (★---- : *DC* breakdown voltage, pressure) fitted using Paschen’s curve with inter-electrode distance as the fitting parameter without cylindrical magnets in the cathode; an experiment performed with three cylindrical Alnico magnets in the cathode, an inter-electrode distance of 0.5 mm; a best fitting parameter of d = 0.56 mm; the value calculated by Equation (A16) was d = 0.53 mm; Gas: He; the magnetic induction was 0.58 T on the surface of the cathode.

**Figure 15 materials-12-00752-f015:**
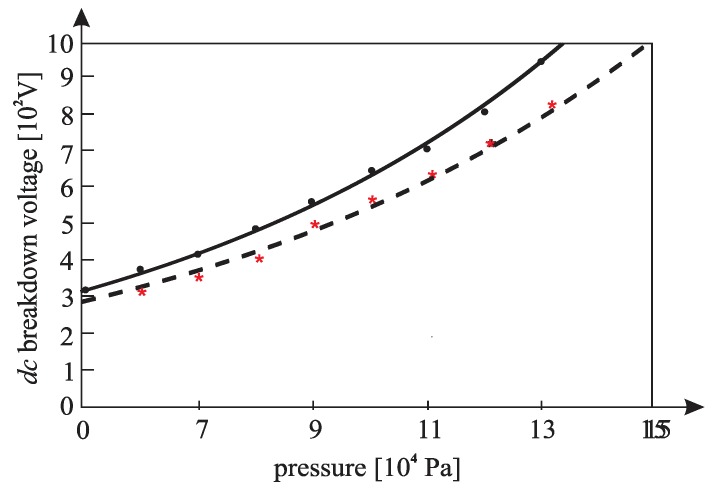
The experimental points of the *DC* breakdown voltage: ― **=** pressure fitted using Paschen’s curve with inter-electrode distance as the fitting parameter; - - - = the theoretical Paschen’s curve determined under the same conditions without cylindrical magnets in the cathode; an experiment performed with three cylindrical Alnico magnets in the cathode, an inter-electrode distance of 0.5 mm; a best fitting parameter of d = 0.55 mm; the value calculated by Equation (A16) was d = 0.52 mm; Gas: He; the magnetic induction was 0.58 T on the surface of the cathode.

**Figure 16 materials-12-00752-f016:**
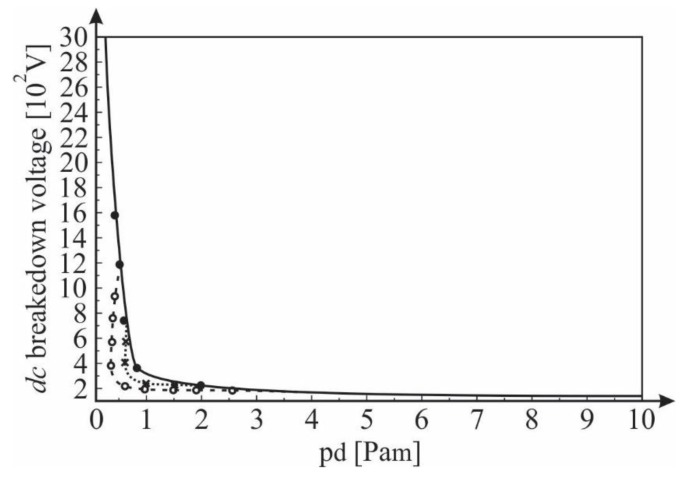
The experimentally obtained values of DC of the breakdowns around Paschen’s minimum: ● cathode with a cavity and ^241^Am, x Rogowski electrode, ○ a Rogowski electrode with a magnetic field; Gas: He; the magnetic induction was 0.58 T on the surface of the cathode.

**Figure 17 materials-12-00752-f017:**
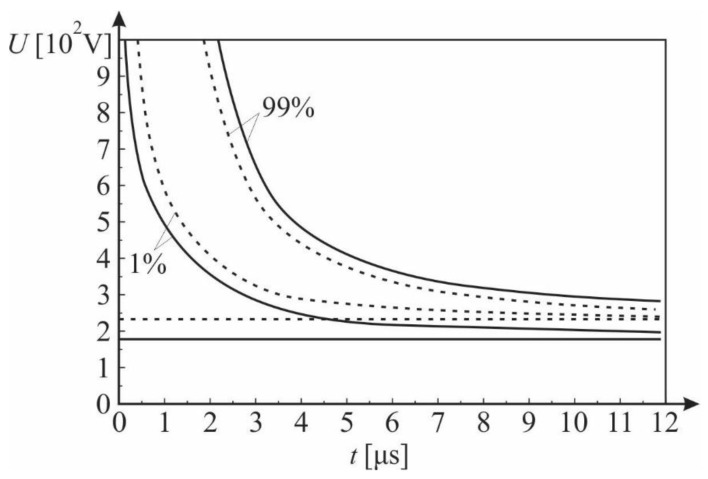
The 99% and 1% impulse characteristics; *PD* = 5 Pa m, ― without a magnetic field, - - - with a magnetic field; Gas: He; the magnetic induction was 0.58 T on the surface of the cathode.

**Table 1 materials-12-00752-t001:** The mean energy of the free electron gas spectrum obtained from different values of the product *PD* and pressure p; ε_1_: the mean energy calculated on the basis of histograms; ε_2_: the mean energy calculated on the basis of Equation (5). ε_3_: the mean energy according to the fitted Maxwell distribution.

PD (Pam)	5	5	105
p (Pa)	5000	50,000	105,000
ε_1_ (eV)	1.85	1.58	1.5
ε_2_ (eV)	1.91	1.97	1.61
ε_3_ (eV)	1.89	1.81	1.68

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
