# Peer review of "The Influence of the Magnetic Field on DC and the Impulse Breakdown of Noble Gases"

_materials, 2019, doi:10.3390/ma12050752_

Reviewer 1 Report

Thank you for your interesting manuscript. Let me make the following comments and requests:
- The introduction is clear and mostly complete except for the references. I am missing references for the statements you make in the second (lines 39-50) and third paragraph of the introduction (51-57).
- Lines 59-62: Room temperature gas discharge are, in many cases, non-Maxwellian (bi-Maxwellian, Druyvesteyn, etc.) depending on the discharge conditions. Hence, I suggest you should explain why assuming Maxwellian distribution is valid in this study.
- Line 68: What does "becoming initial" mean?
- Lines 89-91: I don't think you can state as if this is applicable to noble gases only. As long as EEDF and cross section data is known, the ionization coefficient could be obtained.
- Line 119: "they become initial". Does this mean "they ionize"?
- Lines 139-144: Consider rewording this section. It sounds more complicated than necessary.
- Line 160 "uncertainty type A is 0". I am not familiar with this expression. Please define or reference to the applicable literature.
- Lines 216-218: Consider including a flowchart showing the procedure of the experiment. EEDF (electron energy distribution function) is obtained experimentally for each E/p value -> EEDF is fitted with Eq 2 -> Te is known and alpha is calculated -> Eq 4, 6, 7 is used to calculate the breakdown voltage.
- Line 311: I think this should refer to Eq. 2.
- Fig. 5 top: What does the probability represent? The probability of a breakdown during the impulse tests? If so, why does the probability decrease with increasing electron energy? Shouldn't this be the other way around?
- Fig. 5, in general: Please explain how you obtained each histogram / electron energy. How did you control the electron energy to create each bar of the histogram?
- Fig. 7: Please explain how the electron energy is related to the U[V] values.
- Line 334: Please explain how the electron energy is related to the U[V] values.
- Line 361: Please include the experimental values without magnets for fairer comparison.
- Line 475 "That effect can jeopardize the efficiency of insulation at low voltage level.": I assume what you are trying to say is that the level of protection could be reduced. Or maybe I don't understand what you mean by "efficiency of insulation".
- Fig. A.2.3: Please note the capitalized words in the caption.

Author Response

Reviewer: 1

Summary:
“The introduction is clear and mostly complete except for the references. I am missing references for the statements you make in the second (lines 39-50) and third paragraph of the introduction (51-57).

- Lines 59-62: Room temperature gas discharge are, in many cases, non-Maxwellian (bi-Maxwellian, Druyvesteyn, etc.) depending on the discharge conditions. Hence, I suggest you should explain why assuming Maxwellian distribution is valid in this study.

- Line 68: What does "becoming initial" mean?

- Lines 89-91: I don't think you can state as if this is applicable to noble gases only. As long as EEDF and cross section data is known, the ionization coefficient could be obtained.

- Line 119: "they become initial". Does this mean "they ionize"?

- Lines 139-144: Consider rewording this section. It sounds more complicated than necessary.

- Line 160 "uncertainty type A is 0". I am not familiar with this expression. Please define or reference to the applicable literature.

- Lines 216-218: Consider including a flowchart showing the procedure of the experiment. EEDF (electron energy distribution function) is obtained experimentally for each E/p value -> EEDF is fitted with Eq 2 -> Te is known and alpha is calculated -> Eq 4, 6, 7 is used to calculate the breakdown voltage.

- Line 311: I think this should refer to Eq. 2.

- Fig. 5 top: What does the probability represent? The probability of a breakdown during the impulse tests? If so, why does the probability decrease with increasing electron energy? Shouldn't this be the other way around?

- Fig. 5, in general: Please explain how you obtained each histogram /electron energy. How did you control the electron energy to create each bar of the histogram?

- Fig. 7: Please explain how the electron energy is related to the U[V] values.

- Line 334: Please explain how the electron energy is related to the U[V] values.

- Line 361: Please include the experimental values without magnets for fairer comparison.

- Line 475 "That effect can jeopardize the efficiency of insulation at low voltage level.": I assume what you are trying to say is that the level of protection could be reduced. Or maybe I don't understand what you mean by "efficiency of insulation".

- Fig. A.2..3: Please note the capitalized words in the caption.”

Specific comments:

Q1: The introduction is clear and mostly complete except for the references. I am missing references for the statements you make in the second (lines 39-50) and third paragraph of the introduction (51-57).

A1: Missing references [6,7] and [8,9] has been added (Lines 52 and 59).

6.      Osmokrović, P., Krivokapić, I., Krstić, S., Mechanism of Electrical Breakdown Left of Paschen Minimum, (1994) IEEE Transactions on Dielectrics and Electrical Insulation 1(1), pp. 77-81;

7.      Osmokrović, P., Stability of the gas filled surge arresters characteristics under service conditions, (1996) IEEE Power Engineering Review 16 (1), pp. 54;

8.      Lončar, B., Vujisić, M., Stanković, K., Arandic, D.,Osmokrovic, P., Radioactive resistance of some commercial gas filled surge arresters, 26th International Conference on Microelectronics, MIEL 2008; Nis; Serbia; 11 May 2008 through 14 May 2008;

9.      Vujsić, M., Osmokrović, P., Stanković, K., Lončar, B., Influence of working conditions on over-voltage diode operation, (2007) Journal of Optoelectronics and Advanced Materials, 9 (12), pp. 3881-3884;

Q2: Lines 59-62: Room temperature gas discharge are, in many cases, non-Maxwellian (bi-Maxwellian, Druyvesteyn, etc.) depending on the discharge conditions. Hence, I suggest you should explain why assuming Maxwellian distribution is valid in this study.

A2: Additional explanation has been added as following (Lines 62-80):

“Assuming that the elastic collision frequency is constant, in case when inelastic collisions need not to be taken into consideration, the formation of stable electron spectrum of Maxwell type is thus enabled. This is a consequence of precise balancing of energy gain in the field and elastic energy loss. However, the distribution function of free electron gas f(ε) can be further complicated. The situation is not too intangible. Moreover, in case of large volume filled with weakly ionized monoatomic gases in the weak electrical field (elastic collisions of electron with neutral atom and electron with electron are dominant in respect to inelastic collisions, while the large gas volume ensures the diffusion loss to be small) it can be fully applied. More specifically, the described conditions can be applied at Townsend breakdown of noble gases. In that situation, elastic collisions of electrons with neutral atoms warrant the density of low energy electrons (up to 10 eV) to be rendered significantly higher than the density of high energy electrons. More precisely, in the range of Townsend breakdown, electron-electron collisions can be neglected, since they become dominant when the degree of ionization reaches the values of 10-3 and higher. Yet, these values seem to be very far from Townsend condition of breakdown. This leads to conclusion that f(ε) can be considered Maxwellian in the range of Townsend breakdown. If distribution function tail decreases with function C·exp(-εk), where C is constant and k≈1, this assumption can be considered as a true.”

Q3: Line 68: What does "becoming initial" mean?

A3: Additional explanation about term “becoming initial” has been written in the manuscript, as following (Lines 86-91):

“Becoming initial means that free electron takes energetically favorable state on the free path length and takes enough energy from the electrical field to ionize the neutral Helium atom, creating electron-ion pairs. This newly created electrons can also ionize other neutral Helium atoms and leads to avalanche process, which is actually initiated by the very first electron. Hence, that first electron could be called initial.”

Q4: Lines 89-91: I don't think you can state as if this is applicable to noble gases only. As long as EEDF and cross section data is known, the ionization coefficient could be obtained.

A4: Authors adopted suggestion and part of the text “for noble gases” has been deleted.

Q5: Line 119: "they become initial". Does this mean "they ionize"?

A5: Explanation is given in the answer A3.

Q6: Lines 139-144: Consider rewording this section. It sounds more complicated than necessary.

A6: The section has been reworded, as following (Lines 155-164):

“Given the fact that neither Townsend nor streamer mechanism has inclination to be exclusive, establishing a straightforward boundary between these two mechanisms is not advisable. As a consequence, it is assumed that a broader range of pd product values can be expected to bring about breakdowns thereby combining the two mechanisms. The results of examining the influence exerted by the electrode material and treatment of electrode surface on the static voltage breakdown value lends support to such a supposition. It could be considered that if values of interelectrode gap and mean path length of free electron are approximately comparable, Townsend mechanism of breakdown is dominant. If interelectrode gap length is much higher than mean path length of free electron than streamer mechanism occurs.”

Q7: Line 160 "uncertainty type A is 0". I am not familiar with this expression. Please define or reference to the applicable literature.

A7: In accordance to decision of international metrological institutions in 1993 theory of error analysis was replaced with theory of measurement uncertainty of Type A and Type B. If Type A measurement uncertainty is zero, then deterministic value without statistical deviation is measured.

Corresponding reference [21] has been added at the end of the sentence (Line 178).

Q8: Lines 216-218: Consider including a flowchart showing the procedure of the experiment. EEDF (electron energy distribution function) is obtained experimentally for each E/p value -> EEDF is fitted with Eq 2 -> Te is known and alpha is calculated -> Eq 4, 6, 7 is used to calculate the breakdown voltage.

A8: Corresponding flowchart showing the experimental procedure was added in manuscript shown in Figure 5 (Line 328). Also, part of the text describing the flowchart was added. (Lines 329-348) as following:

    “ Experimental procedure flowchart; 1) Adjustment of chamber operating point without cylindrical magnets in electrodes; 1a) Adjustment of chamber operating point with cylindrical magnets in electrodes; 2) Chamber conditioning; 3) Measuring 1000 values of the impulse breakdown voltage values using 1 kV/s impulses without cylindrical magnets in electrodes; 3a) Measuring 1000 values of the impulse breakdown voltage values using 1 kV/s impulses with cylindrical magnets in electrodes; 4) Determination of the high energy tail distribution of free-electron gas without cylindrical magnets in electrodes; 4a) Determination of the high energy tail distribution of free-electron gas with cylindrical magnets in electrodes; 5) Fitting the high energy tail (with and without cylindrical magnets in electrodes) and drawing obtained results on unique diagram; 6) Determination of the ionization coefficients; 7) Measuring dc breakdown voltage values without cylindrical magnets in electrodes; 7a) Measuring dc breakdown voltage values with cylindrical magnets in electrodes; 8) Drawing Paschen curves based on obtained values for α ionization coefficients and drawing (on the same diagram) measured dc breakdown voltage values (all without cylindrical magnets in electrodes); 8a) Fitting measurement results of dc breakdown voltage versus pd values, using general form of the Paschen curve; 9) The impulse characteristics were determined from results obtained in measurements; 10) Merge the observed results from step 8 and 8a in a unique diagram.”

Q9: Line 311: I think this should refer to Eq. 2.

A9: Corrected.

Q10: Fig. 5 top: What does the probability represent? The probability of a breakdown during the impulse tests? If so, why does the probability decrease with increasing electron energy? Shouldn't this be the other way around?

A10: Probability represent occurring probability of the free high energy electron in the spectrum tail. Detail explanation is given in the answer A11.

Q11: Fig. 5, in general: Please explain how you obtained each histogram /electron energy. How did you control the electron energy to create each bar of the histogram?

A11: For explanation, a part of the text has been added as following (Lines 350-366):

“The free-electron gas spectrum was observed based on 1000 measurements using slow impulse voltage.  More accurate results can be obtained using dc voltage, but in that case, experiment would last much longer. The idea of the experiment was to record impulse breakdown voltage probability histogram. From that histogram, a small probability of breakdown at small voltages ware extracted. Small probability of breakdown achieved by low voltage values [1], microscopically corresponds to the electrons from the tail in the energy spectrum. Those electrons from the tail could take only small amount of energy from electric field (during one free path length) to perform ionization. Hence, knowing the mean path length of free electron and ionization energy of Helium it is possible to draw a tail of the free-electron gas energy histogram. Figures 6a, 7a and 8a show histograms of the class 1 + 3.3•log(n) (n is the size of a statistical sample) of a free electron gas energy distribution and were obtained by fitting the distribution tail with expression C∙exp(-εк). Fitting the tail of the histogram, the value of fitting parameter k≈1 was obtained. Based on that result, it was concluded that the histogram distribution is Maxwellian and the tail of the histogram could be fitted using expression 2. Complete Maxwellian distribution of free electrons is showed on Figures 6b, 7b and 8b.”

Q12: Fig. 7: Please explain how the electron energy is related to the U[V] values.

A12: Explanation is given in the answer A11.

Q13: Line 334: Please explain how the electron energy is related to the U[V] values.

A13: Explanation is given in the answer A11.

Q14: Line 361: Please include the experimental values without magnets for fairer comparison.

A14: Figures 6,7,8,10,11,12,13,14 and 15 has been changed and Figure captions are modified. Also, additional part of the text has been added as discussion as following (Lines 415-418):

“From the Figures 6, 7 and 8 it can be seen that shape of Maxwellian spectrums does not depend on present of magnetic field in interelectrode gap. This is consequence of the fact that magnetic field does not affect kinetic energy of electrons (only rotate them).”

Q15: Line 475 "That effect can jeopardize the efficiency of insulation at low voltage level.": I assume what you are trying to say is that the level of protection could be reduced. Or maybe I don't understand what you mean by "efficiency of insulation".

A15: Additional explanation was added as following (Lines 561-564):

“Apparent increasing of interelectrode distance increase breakdown voltage (nominal voltage). Increasing breakdown voltage leads to shifting of the operating point of GFSA towards higher (U,pd) values.”

Q16: Fig. A.2..3: Please note the capitalized words in the caption.”

A16: Corrected.

Reviewer 2 Report

* Please specify the voltage range encompassed by "low voltage" depending on what part of the world/which international organization you refer to "low voltage" mean entirely different voltage rating.

*line 58 not sure if "precious" should be replaced with "noble" 

* inconsistent spacing between equations and text, not sure if this is something which is resolved by the editor during formatting

* line 189 "Experiment" should it be "Experimental Setup" instead

* line 222-226 for me this is the most important sentence in the paper as it describes the magnetic field which were established, however I found this sentence really confusing to understand. It may make sense to put all of this information into a table so you can easily see the different experiment configurations possible by the electrode setup. This would also make it easier when you are describing the procedure (line 264-271) which also took a while to understand what was going on.

* it is also necessary to state why a magnetic field of 2T would be present in the chamber of the surge arrester. i.e. is there any literature/other research that helped you justify this magnetic field for your experiments?

*Figure 2 - the dimensions should include the units (mm) or mention this in the figure caption

*Extra clarification is required on how the gap distance was set to achieve the desired gap distances especially since you are comparing data collected from various experiment setups. Considering the small difference in experiment data and how small the gap distances are to begin with the argument could also be made that the variation in data collected was due to natural variance when setting up the experiments 

*the dielectric strength of a gas is a functions of it density. In the paper it said all measurements were performed at 0C. can you provide more information about how this temperature was maintained and if it was uniform throughout the pressure vessel. A variation in temperature could effect the values obtained

*While the voltage ramp rates were mentioned no details were provided on the kVA rating of the power supply. This helps to show that the energy released by a breakdown did not cause any localized heating which may affect subsequent measurements.

*On completion of the experiments were the electrodes examined to ensure that breakdown occurred uniformly along its surface. i.e. no breakdown concentration at a certain location which would suggest a non-uniform electric field.

Figures 5-16 should contain information on the gas species (helium) used in the figure caption.

Figure 8 i think the square value should be a gap distance of 1 mm and not 1 m

The strength of the magnetic field should be included in figure captions Figures 9-16

I also noted several small formatting errors with the figures and suggest you read over them again.

Also the units for the figures seemed strange to me. I am used to the breakdown strength being expressed in either V, kV or kV/mm not 10^2 V. Also could the pressure be expressed in kPa instead of 10^3 Pa?

Just as a general recommendation, a colleague of mine Dr Sastry Pamidi has done extensive research over the last few years on characterizing the dielectric strength of helium gas in uniform electric fields. He is interested in using it as a cryogen for superconducting power devices. While the intended applications are different there appears to a good overlap between yours and his research. I would suggest seeing how your experimental data compares to the work he has published

Author Response

Reviewer: 2

Summary:

Line 59-61: Room temperature gas discharge are, in many cases, non-Maxwellian (bi-Maxwellian, Druyvesteyn, etc.) depending on the discharge conditions. Hence, the authors should explain why assuming Maxwellian distribution is valid in this study.

Line 68: What does “becoming initial” mean?

Line 89-90: The authors should not state as if this is applicable to only noble gases. As long as EEDF and cross section data is known, the ionization coefficient could be obtained.

Line 119: What does “they became initial” mean?

Line 139-143: The authors should re-write this section. The message is quite unclear.

Line 160: Please define what uncertainty type A is.

Line 168: Please define what semiempirical surface law is.

Line 216-217: The authors should include a flowchart showing the procedure of the experiment.

Specific comments:

Q1: Room temperature gas discharge are, in many cases, non-Maxwellian (bi-Maxwellian, Druyvesteyn, etc.) depending on the discharge conditions. Hence, the authors should explain why assuming Maxwellian distribution is valid in this study.

A1: Additional explanation has been added as following (Lines 61-80):

“Assuming that the elastic collision frequency is constant, in case when inelastic collisions need not to be taken into consideration, the formation of stable electron spectrum of Maxwell type is thus enabled. This is a consequence of precise balancing of energy gain in the field and elastic energy loss. However, the distribution function of free electron gas f(ε) can be further complicated. The situation is not too intangible. Moreover, in case of large volume filled with weakly ionized monoatomic gases in the weak electrical field (elastic collisions of electron with neutral atom and electron with electron are dominant in respect to inelastic collisions, while the large gas volume ensures the diffusion loss to be small) it can be fully applied. More specifically, the described conditions can be applied at Townsend breakdown of noble gases. In that situation, elastic collisions of electrons with neutral atoms warrant the density of low energy electrons (up to 10 eV) to be rendered significantly higher than the density of high energy electrons. More precisely, in the range of Townsend breakdown, electron-electron collisions can be neglected, since they become dominant when the degree of ionization reaches the values of 10—3 and higher. Yet, these values seem to be very far from Townsend condition of breakdown. This leads to conclusion that f(ε) can be considered Maxwellian in the range of Townsend breakdown. If distribution function tail decreases with function C·exp(εk), where C is constant and k≈1, this assumption can be considered as a true.”

Q2: Line 68: What does “becoming initial” mean?

A2: Additional explanation about term “becoming initial” has been written in the manuscript, as following (Lines 86-91):

“Becoming initial means that free electron takes energetically favorable state on the free path length and takes enough energy from the electrical field to ionize the neutral Helium atom, creating electron-ion pairs. This newly created electrons can also ionize other neutral Helium atoms and leads to avalanche process, which is actually initiated by the very first electron. Hence, that first electron could be called initial.”

Q3: Line 89-90: The authors should not state as if this is applicable to only noble gases. As long as EEDF and cross section data is known, the ionization coefficient could be obtained.

A3: Part of the text “for noble gases” has been deleted from the text.

Q4: Line 119: What does “they became initial” mean?

A4: The explanation has been given in answer A2.

Q5: Line 139-143: The authors should re-write this section. The message is quite unclear.

A5: The section has been reworded, as following (Lines 155-164):

“Given the fact that neither Townsend nor streamer mechanism has inclination to be exclusive, establishing a straightforward boundary between these two mechanisms is not advisable. As a consequence, it is assumed that a broader range of pd product values can be expected to bring about breakdowns thereby combining the two mechanisms. The results of examining the influence exerted by the electrode material and treatment of electrode surface on the static voltage breakdown value lends support to such a supposition. It could be considered that if values of interelectrode gap and mean path length of free electron are approximately comparable, Townsend mechanism of breakdown is dominant. If interelectrode gap length is much higher than mean path length of free electron than streamer mechanism occurs.”

Q6: Line 160: Please define what uncertainty type A is.

A6: In accordance to decision of international metrological institutions in 1993 theory of error analysis was replaced with theory of measurement uncertainty of Type A and Type B. If Type A measurement uncertainty is zero, then deterministic value without statistical deviation is measured.

Corresponding reference has been added at the end of the sentence (Line 178).

Q7: Line 168: Please define what semiempirical surface law is.

A7: Additional explanation authors gave in the Appendix 1. Please, see Appendix 1.

Q8: Line 216-217: The authors should include a flowchart showing the procedure of the experiment.

A8: Corresponding flowchart showing the experimental procedure was added in manuscript (Line 328). Also, part of the text, describing the flowchart, was added (Lines 329-348) as following:

     “ Experimental procedure flowchart; 1) Adjustment of chamber operating point without cylindrical magnets in electrodes; 1a) Adjustment of chamber operating point with cylindrical magnets in electrodes; 2) Chamber conditioning; 3) Measuring 1000 values of the impulse breakdown voltage values using 1 kV/s impulses without cylindrical magnets in electrodes; 3a) Measuring 1000 values of the impulse breakdown voltage values using 1 kV/s impulses with cylindrical magnets in electrodes; 4) Determination of the high energy tail distribution of free-electron gas without cylindrical magnets in electrodes; 4a) Determination of the high energy tail distribution of free-electron gas with cylindrical magnets in electrodes; 5) Fitting the high energy tail (with and without cylindrical magnets in electrodes) and drawing obtained results on unique diagram; 6) Determination of the ionization coefficients; 7) Measuring dc breakdown voltage values without cylindrical magnets in electrodes; 7a) Measuring dc breakdown voltage values with cylindrical magnets in electrodes; 8) Drawing Paschen curves based on obtained values for α ionization coefficients and drawing (on the same diagram) measured dc breakdown voltage values (all without cylindrical magnets in electrodes); 8a) Fitting measurement results of dc breakdown voltage versus pd values, using general form of the Paschen curve; 9) The impulse characteristics were determined from results obtained in measurements; 10) Merge the observed results from step 8 and 8a in a unique diagram.”

Reviewer 3 Report

Overview:

This paper could be summarized as follows:

EEDF (electron energy distribution function) is obtained experimentally for each E/p value -> EEDF is fitted with Eq 2 -> Now Te is known and alpha is calculated -> Eq 4, 6, 7 is used to calculate the breakdown voltage. This procedure is taken with and without magnetic field to see the magnetic field influence.

Comments for the authors:

Line 59-61: Room temperature gas discharge are, in many cases, non-Maxwellian (bi-Maxwellian, Druyvesteyn, etc.) depending on the discharge conditions. Hence, the authors should explain why assuming Maxwellian distribution is valid in this study.

Line 68: What does “becoming initial” mean?

Line 89-90: The authors should not state as if this is applicable to only noble gases. As long as EEDF and cross section data is known, the ionization coefficient could be obtained.

Line 119: What does “they become initial” mean?

Line 139-143: The authors should re-write this section. The message is quite unclear.

Line 160: Please define what uncertainty type A is.

Line 168: Please define what semiempirical surface law is.

Line 216-217: The authors should include a flowchart showing the procedure of the experiment.

EEDF (electron energy distribution function) is obtained experimentally for each E/p value -> EEDF is fitted with Eq 2 -> Te is known and alpha is calculated -> Eq 4, 6, 7 is used to calculate the breakdown voltage.

Line 311: Eq.3 shouldn’t this be Eq.2?

Figure 5a: What does the probability represent? The probability of a breakdown during the impulse tests? If so, why does the probability decrease with increasing electron energy? Shouldn't this be the other way around?

Figure 5b: The authors should explain how they obtained each histogram / electron energy. How did the authors control the electron energy to create each bar of the histogram?

Figure 7b: The authors should explain how the electron energy is related to the U[V] values.

Line 334: Based on Figures 5,6,7: The authors should also plot the measured EEDF (electron energy distribution function) when B field is applied for comparison.

Line 361: Figures 9, 10, 11, 12, 13, and 14: The authors should include the experimental values without magnets for fairer comparison.

Author Response

Reviewer: 3

Summary:

*Please specify the voltage range encompassed by "low voltage" depending

on what part of the world/which international organization you refer to

"low voltage" mean entirely different voltage rating.

*line 58 not sure if "precious" should be replaced with "noble"

* inconsistent spacing between equations and text, not sure if this is something which is resolved by the editor during formatting

* line 189 "Experiment" should it be "Experimental Setup" instead

* line 222-226 for me this is the most important sentence in the paper as it describes the magnetic field which were established, however I found this sentence really confusing to understand. It may make sense to put all of this information into a table so you can easily see the different experiment configurations possible by the electrode setup. This would also make it easier when you are describing the procedure (line 264-271) which also took a while to understand what was going on.

* it is also necessary to state why a magnetic field of 2T would be present in the chamber of the surge arrester. i.e. is there any literature/other research that helped you justify this magnetic field for your experiments?

*Figure 2 - the dimensions should include the units (mm) or mention this in the figure caption

*Extra clarification is required on how the gap distance was set to achieve the desired gap distances especially since you are comparing data collected from various experiment setups. Considering the small difference in experiment data and how small the gap distances are to begin with the argument could also be made that the variation in data collected was due to natural variance when setting up the experiments

*the dielectric strength of a gas is a function of its density. In the paper it said all measurements were performed at 0C. can you provide more information about how this temperature was maintained and if it was uniform throughout the pressure vessel. A variation in temperature could affect the values obtained

*While the voltage ramp rates were mentioned no details were provided on the kVA rating of the power supply. This helps to show that the energy released by a breakdown did not cause any localized heating which may affect subsequent measurements.

*On completion of the experiments were the electrodes examined to ensure that breakdown occurred uniformly along its surface. i.e. no breakdown concentration at a certain location which would suggest a non-uniform electric field.

*Figures 5-16 should contain information on the gas species (helium) used in the figure caption.

*Figure 8 i think the square value should be a gap distance of 1 mm and not 1 m

*The strength of the magnetic field should be included in figure captions Figures 9-16

*I also noted several small formatting errors with the figures and suggest you read over them again.

* Also, the units for the figures seemed strange to me. I am used to the breakdown strength being expressed in either V, kV or kV/mm not 10^2 V. Also, could the pressure be expressed in kPa instead of 10^3 Pa?

Specific comments:

Q1:Please specify the voltage range encompassed by "low voltage" depending
on what part of the world/which international organization you refer to
"low voltage" mean entirely different voltage rating.

A1: Corresponding reference was added as following (Line 36):

„...The most commonly used low-voltage [1] overvoltage protection component is a gas surge arrester (in German literature known as a fuse with a noble gas) [2,3,4].

1.       IEEE C62.31-2006 - IEEE Standard Test Methods for Low-Voltage Gas-Tube Surge-Protective Device Components.

Q2: line 58 not sure if "precious" should be replaced with "noble"

A2: Word "precious" has been replaced with "noble"

Q3: Inconsistent spacing between equations and text, not sure if this is something which is resolved by the editor during formatting

A3: Corrected.

Q4: line 189 "Experiment" should it be "Experimental Setup" instead

A4: Corrected.

Q5: line 222-226 for me this is the most important sentence in the paper as it describes the magnetic field which were established, however I found this sentence really confusing to understand. It may make sense to put all of this information into a table so you can easily see the different experiment configurations possible by the electrode setup. This would also make it easier when you are describing the procedure (line 264-271) which also took a while to understand what was going on.

A5: Additional explanation has been added in text as following (Lines 236-239):

“One chamber was designed to operate in underpressure regime/mode (pressure in the chamber), and the other in overpressure (pressure in the chamber). Difference between these two chambers was in O-ring channel profile. The electrodes of the axially symmetric system made of copper were in the form of the Rogowski (to avoid electric field edge effect).”

Q6: It is also necessary to state why a magnetic field of 2T would be present in the chamber of the surge arrester. i.e. is there any literature/other research that helped you justify this magnetic field for

your experiments?

A6: There was an error in text regarding to value of magnetic induction. Value of magnetic induction is 1,2 T (one point two Tesla). So, part of the text “Br = 1, 2T” replaced with “Br = 1,2 T”

Additionally, explanations were added in text as following (Lines 48-50):

“It has been shown that strong magnetic field stabilize operating point and decrease statistical dispersion of the breakdown voltage in two-electrode system [5, 6].”

And also (Lines 243-246) as following:

“In one of the electrodes (commonly used as a cathode) it was possible to install 1, 2 or 3 identical cylindrical alnico magnets of induction Br = 1,2 T (cylindrical magnets with the strongest induction value that was available for experiment), with a radius R = 10 mm and height N = 10 mm (shown as detail in Figure 2). These magnets created magnetic field collinear with electric field in the interelectrode space, induction 0,58 T on the surface of the cathode (Appendix 3)”.

Q7: Figure 2 - the dimensions should include the units (mm) or mention this in the figure caption

A7: Figure 2 caption was supplemented with text “The units are in mm.”

Q8: Extra clarification is required on how the gap distance was set to achieve the desired gap distances especially since you are comparing data collected from various experiment setups. Considering the small difference in experiment data and how small the gap distances are to begin with the argument could also be made that the variation in data collected was due to natural variance when setting up the experiments.

A8: Clarification was added (Lines 256-258) as following:

“The Type B measurement uncertainty of interelectrode distance adjustment in this procedure was 0,2 %. This result is confirmed by measuring 1000 values of dc breakdown voltages (Type A measurement uncertainty was zero).”

Also, significant error in measurement results was not possible to get, because it would be rejected by statistical analyses (U test, and Shauvenet’s criteria) that were performed.

Q9: the dielectric strength of a gas is a function of it density. In the paper it said all measurements were performed at 0C. can you provide more information about how this temperature was maintained and if it was uniform throughout the pressure vessel. A variation in temperature could

effect the values obtained

A9:  In accordance to Gay-Lussac law value of pressure p(T) at the temperature T is related to pressure p(0) at temperature T =0°C according to expression:

p(0) = (1+ T / 273,15)-1 Ÿp(T)

Also, part of the text was added in the manuscript (Line 263) as following:

“set the desired pressure value to the chamber reduced to 0° C by using Gay-Lussac law.”

Q10: While the voltage ramp rates were mentioned no details were provided on the kVA rating of the power supply. This helps to show that the energy released by a breakdown did not cause any localized heating which may affect subsequent measurements

A10: Additional information has been added in text as following (Lines 275-278):

“The chamber (cathode) was grounded using 1 MΩ resistor which secured negligible irreversible changes in topography of the electrode surface during measurement. The success of this measure was confirmed by talysurf before and after one series of measurements.”

Q11:  On completion of the experiments were the electrodes examined to ensure that breakdown occurred uniformly along its surface. i.e. no breakdown concentration at a certain location which would suggest a non-uniform electric field.

A11: The absence of electric field non-uniformity was assured using Rogowski type of electrode.

Also, additional sentence has been added in text as following (Lines 278-280):

“Examining the electrode surfaces by microscope it was concluded that breakdown points were uniformly distributed along electrode surfaces.”
Q12:  Figures 5-16 should contain information on the gas species (helium) used in the figure caption.

A12: Figure captions were supplemented with gas information.

Q13:  Figure 8 i think the square value should be a gap distance of 1 mm and not 1 m

A13: Corrected.

Q14: The strength of the magnetic field should be included in figure captions
Figures 9-16

A14: Corrected.

Q15: also noted several small formatting errors with the figures and suggest you read over them again.

A15: Corrected.

Q16: Also, the units for the figures seemed strange to me. I am used to the breakdown strength being expressed in either V, kV or kV/mm not 10^2 V. Also, could the pressure be expressed in kPa instead of 10^3 Pa?

A16: Authors generally accept comment and are very appreciate, but the deadline for manuscript revision is too short (had three referees with very useful comments). Also, authors will be grateful to get additional references on Dr Pamidi work, and would like to change experience in field of our research work.

Round  2

Reviewer 3 Report

Dear Authors:

Thank you for addressing most of my comments.

However, I do have some follow up questions and requests.

Line 414-416: The authors stated that “From the Figures 6, 7 and 8 it can be seen that shape of Maxwellian spectrums does not depend on present of magnetic field in interelectrode gap.”: if this is the case, what is causing the difference in dc breakdown voltage values shown in Figures 10–15?

Line 414-416: According to the equation of single particle motion, the presence of B field changes the acceleration of electrons (example: if electrons were traveling in x-axis direction, B field will apply force in y-axis direction: additional force will change the x, y velocity component). Therefore, I think the following statement is incorrect: “This is consequence of the fact that magnetic field does not affect kinetic energy of electrons (only rotate them)”. Wouldn't the addition of rotational motion change the kinetic energy of the electrons? Please validate this statement.

F(e) in Figures 6, 7 and 8 show lower or higher energy tails when there is B field, however, Figures 10, 11 and 12 all show lower dc breakdown voltage values with the presence of B field. I had some difficulties understanding the correlation among these figures. Could you please clarify why this is the case?

Thank you.

Author Response

Title: The influence of the magnetic field on dc and the impulse breakdown of noble gases

Journal: Materials 

Dear Editor, 

Below are our answers to additional questions from second round reviewer, marked orange.

Best regards,

Authors

Second round reviewer: 

Summary:

“Dear Authors: 

Thank you for addressing most of my comments.

However, I do have some follow up questions and requests.

Line 414-416: The authors stated that “From the Figures 6, 7 and 8 it can be seen that shape of Maxwellian spectrums does not depend on present of magnetic field in interelectrode gap.”: if this is the case, what is causing the difference in dc breakdown voltage values shown in Figures 10–15?

Line 414-416: According to the equation of single particle motion, the presence of B field changes the acceleration of electrons (example: if electrons were traveling in x-axis direction, B field will apply force in y-axis direction: additional force will change the x, y velocity component). Therefore, I think the following statement is incorrect: “This is consequence of the fact that magnetic field does not affect kinetic energy of electrons (only rotate them)”. Wouldn't the addition of rotational motion change the kinetic energy of the electrons? Please validate this statement.

F(e) in Figures 6, 7 and 8 show lower or higher energy tails when there is B field, however, Figures 10, 11 and 12 all show lower dc breakdown voltage values with the presence of B field. I had some difficulties understanding the correlation among these figures. Could you please clarify why this is the case?

Thank you.”

Specific comments:

Q1 and Q2:

Line 414-416: The authors stated that “From the Figures 6, 7 and 8 it can be seen that shape of Maxwellian spectrums does not depend on present of magnetic field in interelectrode gap.”: if this is the case, what is causing the difference in dc breakdown voltage values shown in Figures 10–15?

Line 414-416: According to the equation of single particle motion, the presence of B field changes the acceleration of electrons (example: if electrons were traveling in x-axis direction, B field will apply force in y-axis direction: additional force will change the x, y velocity component). Therefore, I think the following statement is incorrect: “This is consequence of the fact that magnetic field does not affect kinetic energy of electrons (only rotate them)”. Wouldn't the addition of rotational motion change the kinetic energy of the electrons? Please validate this statement.

A1 and A2: Authors are grateful for particular comments (Q1 and Q2). The text has been edited and authors have written a unifying explanation to Q1 and Q2 as following (Lines 406-427):

„Based on results (Figures 6, 7, 8 and Table 1) it can be concluded that the free electron gas spectrum in the noble gas helium (He) is of the Maxwellian shape. Also, that the overall energy of the initial electron (increased for the amount of rotation energy around vector B does not affect Maxwellian shape of free electron gas spectrum. The latter is due to the existence of an elastic type of collision between the He atoms and free electrons. Such a state of thermodynamic equilibrium is neither disturbed by an electric nor a magnetic field. A smaller deviation of the free electron gas spectrum occurs at higher pressure values, as a result of the appearance of the inelastic Coulomb, electron-electron interactions. Although shown in Figures 6,7,8 and Table 1, this phenomenon does not significantly disturb a shape of the free electron gas spectrum (it only leads to an increase in the spectrum density at lower energies).

As already stated, cross section σi(ε) is practically independent of the free electron energy and v is the velocity in the direction of the electric field independent from magnetic field B.  In addition, and consistent with the results given in Figures 6, 7 and 8, the shape of Maxwellian spectrum does not depend on the presence of magnetic field in interelectrode gap. Thus, Eq.1 allows for the determination of the ionization coefficient. This means that magnetic field has negligible influence on ionization (avalanche) process, but does affect the value of breakdown voltage because the superposition of translational and rotational motion of initial electron leads to an increase of its path length i.e. to an apparent increase in the interelectrode distance (Appendix 2).”  

Q3: F(e) in Figures 6, 7 and 8 show lower or higher energy tails when there is B field, however, Figures 10, 11 and 12 all show lower dc breakdown voltage values with the presence of B field. I had some difficulties understanding the correlation among these figures. Could you please clarify why this is the case?

A3: Additional clarification has been written in the manuscript, as following (Lines 510-516):

“Consequently, the values of dc breakdown voltage at the points to the left of Paschen minimum (Figures 10, 11 and 12) are lower when there exists magnetic field in the inter-electrode gap and vice versa, at the points to the right of Paschen minimum (Figures 13, 14 and 15). The latter is attributable to the anomalous Paschen effect, which appears at the points to the left of Paschen minimum [28]. According to the anomalous Paschen effect, lower values of breakdown voltage correspond to higher inter-electrode distances.”
